# *WBP5* Expression Influences Prognosis and Treatment Response in Head and Neck Squamous Cell Carcinoma

**DOI:** 10.3390/cancers17040587

**Published:** 2025-02-08

**Authors:** Eun-jeong Jeong, Eunjeong Kim, Kwang-Yoon Jung, Seung-Kuk Baek, Yeon Soo Kim

**Affiliations:** 1Department of Otorhinolaryngology–Head and Neck Surgery, Konyang University College of Medicine, Daejeon 35365, Republic of Korea; 602547@kyuh.ac.kr; 2Department of Otorhinolaryngology–Head and Neck Surgery, Korea University College of Medicine, Seoul 02841, Republic of Korea; kyjung@korea.ac.kr (K.-Y.J.); mdbsk@korea.ac.kr (S.-K.B.); 3BK21 FOUR KNU Creative BioResearch Group, Department of Biology, College of Natural Sciences, Kyungpook National University, Daegu 41566, Republic of Korea; eunjkim@knu.ac.kr

**Keywords:** head and neck squamous cell carcinoma, WW domain-binding protein 5 (WBP5), Transcriptional Elongation Factor A-like 9 (TCEAL9)

## Abstract

Head and neck squamous cell carcinoma (HNSCC) exhibits complex genetic alterations, and *WBP5* has been identified as a potential therapeutic target. A comprehensive bioinformatics analysis of public datasets revealed that *WBP5* is overexpressed in HNSCC and is associated with poor prognosis. Silencing *WBP5* in HNSCC cells reduced cell proliferation and viability while increasing sensitivity to cisplatin, suggesting its potential involvement in chemoresistance. These findings highlight the potential of *WBP5* as a prognostic biomarker and therapeutic target. Modulating *WBP5* expression may improve treatment outcomes in HNSCC, and further research is needed to elucidate its mechanisms and develop targeted therapies.

## 1. Introduction

Head and neck squamous cell carcinoma (HNSCC) represents a prevalent and highly aggressive malignancy originating from the squamous epithelial cells that constitute the mucosal linings of the head and neck region, encompassing the oral cavity, pharynx, and larynx. The etiology of HNSCC is strongly linked to tobacco use, chronic alcohol consumption, and human papillomavirus (HPV) infection. Additionally, genetic and epigenetic alterations, including mutations in TP53, CDKN2A, and aberrant signaling in pathways such as EGFR, PI3K/AKT, and NOTCH, play crucial roles in its pathogenesis. Current therapeutic paradigms for HNSCC incorporate surgical resection, radiotherapy, chemotherapy, or multimodal approaches combining these treatments. However, the persistently low 5-year survival rate of approximately 50% among treated patients underscores an urgent need to develop advanced and more efficacious therapeutic interventions [1,2].

WW domain protein 5 (*WBP5*), alternatively referred to as Transcriptional Elongation Factor A-like 9 (*TCEAL9*), engages with diverse signaling pathways and associated with WW domains [3]. WW domains are protein–protein interaction motifs that recognize and bind to short proline-rich sequences. Although the biological role of WBP5 is remains unclear, its expression and functional relevance in diseases, particularly cancer, have been increasingly recognized.

Sanger sequencing has identified mutations in *WBP5* on the X chromosome in patients with colorectal cancer with microsatellite instability, implicating it as a potential mutational driver along with 14 other genes, including *ADAR*, *DCAF12L2*, *GLT1D1*, *ITGA7*, *MAP1B*, *MRGPRX4*, *PSRC1*, *RANBP2*, *RPS6KL1*, *SNCAIP*, *TCEAL6*, *TUB6*, *VEGFB*, and *ZBTB2* [4,5]. In small-cell lung cancer, WBP5 overexpression has been linked to drug resistance, mediated by microRNA-335 via regulation of the Hippo pathway [4]. Additionally, in acute myeloid leukemia (AML), *WBP5* overexpression has been associated with poor prognosis, correlating with increased expression of *HOX* gene cluster (*HOXA5*, *HOXA7*, *HOXA9*, and *HOXA10*), *MEIS1*, and *FOXC1* [3]. These findings highlight the emerging significance of WBP5 in cancer biology, suggesting its potential role in tumor progression, drug resistance, and poor clinical outcomes across different cancer types.

The role of *WBP5* in the pathogenesis of HNSCC remains poorly characterized. In this study, we performed an in-depth analysis of publicly available datasets and observed a significant overexpression of WBP5 in HNSCC patients. Notably, elevated WBP5 expression was strongly associated with unfavorable clinical outcomes, including reduced overall survival and increased metastatic potential. Functional in vitro experiments using HNSCC cell lines revealed that WBP5 plays a critical role in driving tumor cell proliferation and mediating chemoresistance. These findings establish WBP5 as a potential prognostic biomarker and therapeutic target for the treatment of HNSCC.

## 2. Results

### 2.1. Differential and Validated Expression of WBP5 in Various Cancer Types

The GEPIA2 database was utilized to evaluate *WBP5* expression levels across various cancer types, revealing a significant upregulation of WBP5 in tumor tissues compared to normal tissues in several cancers, including HNSCC (Figure 1A). Specifically, in HNSCC, WBP5 expression was significantly higher in tumor samples (*n* = 519) than in normal tissues (*n* = 44) (Figure 1B) (*p* < 0.05). These findings suggest that WBP5 may play a crucial role in tumorigenesis and serve as a potential biomarker for cancer. To confirm these results, *WBP5* mRNA expression levels were analyzed in independent datasets from the Gene Expression Omnibus (GEO). Consistent with TCGA data, *WBP5* expression was significantly upregulated in tumor tissues compared to normal tissues across seven independent datasets, including GSE58911, GSE178537, and GSE32332 (*p* < 0.05) (Figure 1C–E).

The robust consistency of these results across both the TCGA and GEO datasets further underscores the possible oncogenic role of *WBP5* in cancer progression, highlighting its potential as a critical molecular target in tumor biology.

### 2.2. Prognostic Value of WBP5 Expression in HNSC

The prognostic relevance of *WBP5* expression in HNSCC was thoroughly evaluated through Kaplan–Meier survival analyses, revealing a strong association between elevated *WBP5* expression and poor clinical outcomes. Using the GEPIA2 database, we analyzed survival data for HNSCC and other cancers. The survival map generated from GEPIA2 indicated that higher *WBP5* expression levels were significantly associated with reduced survival rates in patients with HNSCC and liver hepatocellular carcinoma (LIHC) (Figure 2A). Specifically, Kaplan–Meier analyses of overall survival (OS) demonstrated that elevated *WBP5* expression was linked to significantly worse survival outcomes (hazard ratio [HR] = 0.0015, *p* = 0.0012) (Figure 2B). To validate these findings, we conducted additional Kaplan–Meier survival analyses, which were consistent with the GEPIA2 results. Patients with high *WBP5* expression exhibited significantly reduced OS (HR = 1.75, 95% CI: 1.25–2.45, log-rank *p* = 0.00089) and relapse-free survival (RFS) (HR = 2.17, 95% CI: 0.98–4.8, log-rank *p* = 0.05) (Figure 2C). These findings underscore the significant role of *WBP5* in negatively influencing both OS and RFS outcomes in HNSCC. Further stratified analyses examined the relationship between *WBP5* expression and survival outcomes across different tumor grades. In grade I tumors, patients with high *WBP5* expression had a substantially higher risk of mortality compared to those with low *WBP5* expression (HR = 3.55, 95% CI: 1.17–10.75, log-rank *p* = 0.017) (Figure 3A). This suggests that *WBP5* may have a critical role in driving early-stage tumor progression. For grade II tumors, high *WBP5* expression levels were also significantly associated with reduced survival rates (HR = 1.41, 95% CI: 1–2, log-rank *p* = 0.05) (Figure 3B). Similarly, in advanced grade III tumors, elevated *WBP5* levels were predictive of poorer survival outcomes (HR = 2.1, 95% CI: 1.02–4.3, log-rank *p* = 0.038) (Figure 3C). 

These consistent trends across tumor grades highlight *WBP5*’s pivotal role in promoting tumor progression and its prognostic value in HNSCC. In addition, sex-stratified analyses were performed to explore the influence of *WBP5* expression on survival outcomes in male and female patients with HNSCC (Appendix A). Among male patients, those with high *WBP5* expression levels exhibited significantly worse overall survival compared to those with low *WBP5* expression (HR = 1.63, 95% CI: 1.15–2.32, log-rank *p* = 0.0054) (Appendix A). Similarly, in female patients, high *WBP5* expression was also associated with reduced survival (HR = 1.9, 95% CI: 1.06–3.43, log-rank *p* = 0.029) (Appendix A). These results demonstrate that elevated *WBP5* expression negatively impacts survival outcomes across sexes, further supporting its role as a robust prognostic biomarker in HNSCC. Taken together, these findings strongly suggest that *WBP5* expression serves as a reliable prognostic biomarker across various clinical and pathological parameters in HNSCC. The consistent association between high *WBP5* expression and poor survival outcomes underscores its potential role in guiding personalized treatment strategies for HNSCC patients. Further studies are warranted to explore the underlying mechanisms by which *WBP5* contributes to tumor progression and poor prognosis, and to assess its utility as a therapeutic target.

### 2.3. Association of WBP5 with Immune Cell Infiltration

The interplay between *WBP5* expression and immune cell infiltration was evaluated to elucidate its potential role within the tumor microenvironment and its influence on immune-mediated mechanisms. Immune cell infiltration is a critical determinant of tumor progression, immune escape, and clinical outcomes, as infiltrating immune cells can exhibit either tumor-suppressive or tumor-promoting functions depending on their subtype and activation state.

Deciphering the extent to which *WBP5* expression modulates or correlates with the infiltration of specific immune cell populations is essential for understanding its contribution to the tumor–immune system landscape and its potential utility as a therapeutic target in immunotherapy. Elevated *WBP5* expression levels were strongly associated with significantly worse survival outcomes in patients exhibiting either increased or decreased levels of key immune cell subsets, including natural killer T (NKT) cells, regulatory T cells, CD4+ memory T cells, and macrophages (Figure 4). A high *WBP5* expression level was associated with an increased risk of mortality in patients with enriched (HR = 2.54, log-rank *p* = 0.0052) and decreased NKT cells (HR = 1.66, log-rank *p* = 0.0096) (Figure 4A). Therefore, increased *WBP5* expression levels could hinder the antitumor function of NKT cells, which are essential for identifying and destroying malignant cells.

A similar association was observed in regulatory T cells. In patients with enriched regulatory T cell populations, high *WBP5* expression levels predicted poor survival (HR = 1.53, log-rank *p* = 0.044), whereas in those with decreased regulatory T cells, the HR was even higher (HR = 2.53, log-rank *p* = 0.0012). Thus, *WBP5* may influence the immunosuppressive functions of regulatory T cells, thereby contributing to tumor immune evasion (Figure 4B). The relationship between *WBP5* expression and CD4+ memory T cells was examined. A high *WBP5* expression level was associated with worse outcomes in patients with both enriched (HR = 1.86, log-rank *p* = 0.0048) and decreased (HR = 2.05, log-rank *p* = 0.0025) CD4+ memory T cell levels, highlighting its broad impact on adaptive immune responses (Figure 4C). In cancer, macrophages exhibit dual characteristics, functioning as either pro-tumorigenic (M2-like) or antitumorigenic (M1-like) entities. When *WBP5* was overexpressed, patient survival was reduced regardless of the presence or absence of macrophages. Thus, *WBP5* may impair the function of anti-tumorigenic macrophages, potentially exacerbating tumor progression (Figure 4D). 

Overall, these findings underscore the multifaceted role of *WBP5* in shaping the tumor immune microenvironment and its potential involvement in immune evasion. By influencing the infiltration and functionality of critical immune cell subsets, *WBP5* may contribute to an immunosuppressive microenvironment that promotes tumor growth and resistance to immune-mediated therapies. This highlights the importance of further investigating *WBP5* as a potential target for combination therapies aimed at enhancing antitumor immunity. 

### 2.4. Relationship of WBP5 Expression with Tumor Grade, Stage, and Metastasis

Using a tumor–node–metastasis (TNM) plot analysis, we investigated the association between *WBP5* expression and critical clinical parameters, including tumor stage, tumor grade, and nodal metastasis status, to better understand its role in the progression and aggressiveness of head and neck squamous cell carcinoma (HNSCC).

First, the analysis of *WBP5* expression across tumor stages revealed significant differences between normal tissues and cancerous tissues, with a marked increase observed in tumors (Figure 5A). While the expression did not show a strictly linear progression across all stages, it remained consistently elevated in advanced stages (Stage III and IV) compared to early-stage cancers (Stage I and II). This pattern suggests that *WBP5* plays an important role in tumor initiation and continues to influence tumor progression. WBP5 expression was evaluated based on tumor grade, revealing a strong association with tumor differentiation status (Figure 5B). High-grade tumors, characterized by poorly differentiated and more aggressive cancer cells, exhibited significantly higher *WBP5* expression levels compared to low-grade tumors and normal tissues. Although the differences between intermediate grades were less pronounced, the elevated expression in high-grade tumors highlights *WBP5*’s role in promoting aggressive tumor phenotypes. Lastly, we examined the relationship between *WBP5* expression and nodal metastasis to assess its potential involvement in metastatic progression (Figure 5C). Tumors with nodal metastasis (N1–N3) showed significantly higher *WBP5* expression levels compared to those without metastasis (N0) and normal tissues. Although expression levels did not significantly vary among metastatic groups (N1–N3), the overall elevation in metastatic cases indicates *WBP5*’s association with the metastatic phenotype, potentially contributing to the spread of HNSCC.

Taken together, these results demonstrate significant correlations between increased *WBP5* expression and advanced tumor stages, higher tumor grades, and nodal metastasis. Although the expression patterns are not strictly dose-dependent, *WBP5* appears to play a critical role in tumor initiation, progression, and the development of aggressive and metastatic phenotypes. These findings position *WBP5* as a promising prognostic biomarker and a potential therapeutic target for patients with advanced or high-risk HNSCC. Further studies are warranted to elucidate the molecular mechanisms underlying *WBP5*’s role in tumor progression and to explore its utility in therapeutic interventions.

### 2.5. Correlation Between WBP5 and Epidermal Growth Factor Receptor (EGFR) Expression

The *EGFR* is essential for controlling cell proliferation, survival, angiogenesis, and metastasis. It is often overexpressed in HNSCC, contributing to tumor aggressiveness and resistance to therapy. To explore their potential interactions, we investigated the correlation between *WBP5* and *EGFR* expression in HNSCC.

We examined the correlation between *WBP5* and *EGFR* expressions in various cancer types (Appendix A). *WBP5* was overexpressed in multiple cancers, including CHOL, DLBC, ESCA, LGG, LIHC, PAAD, SARC, SKCM, STAD, THYM, UCS, and HNSCC, compared to those in normal samples. Particularly, in HNSCC, both *WBP5* and *EGFR* expressions were significantly upregulated in tumor tissues compared to those in normal samples (Figure 6A). A scatter plot analysis further confirmed the significant positive correlation between *WBP5* and *EGFR* expressions in HNSCC (Figure 6B). Additional validation using the GSE33232 and GSE178537 datasets demonstrated a statistically significant positive correlation between *WBP5* and *EGFR* expressions (Figure 6C). These findings suggest that *WBP5* and *EGFR* may act as potential coregulators in HNSCC, highlighting the possible role of *WBP5* as a therapeutic target in this cancer type.

### 2.6. Functional Validation of WBP5 in Tumorigenesis

siRNA-mediated knockdown experiments were conducted in patients with HNSCC to assess the functional role of *WBP5* and examine its impact on cell proliferation, drug sensitivity, and tumor progression.

The knockdown of *WBP5* using siRNA resulted in a significant reduction in *WBP5* expression, as confirmed using Quantitative real-time PCR (qRT-PCR) analysis (Figure 7A). Reduction in *WBP5* expression significantly inhibited the proliferation of FaDu cells, as demonstrated by cell growth assays (Figure 7B). Additionally, *WBP5*-knockdown FaDu cell exhibited increased sensitivity to cisplatin, a chemotherapeutic agent commonly used for head and neck cancers (Figure 7C). An assessment of capase-3/7 activity demonstrated that silencing *WBP5* resulted in a marked elevated of caspase-3/7 activity, suggesting increased apoptotic activity in cells lacking *WBP5* expression (Figure 7D).

Finally, clonogenic formation assays showed that WBP5-knockdown FaDu cells exhibited a significant reduction in colony formation ability, which was further diminished by cisplatin treatment (Figure 7E). Thus, *WBP5* may contribute to cell proliferation, drug sensitivity, and tumor formation in HNSCC. Targeting WBP5 may enhance the efficacy of chemotherapy and inhibit tumor progression, highlighting its potential as a therapeutic target for HNSCC.

## 3. Discussion

Our findings demonstrate that *WBP5* is highly overexpressed in HNSCC and is associated with poorer survival rates and increased metastatic potential. Functional studies on HNSCC cell lines confirmed that *WBP5* modulates cell proliferation and enhances resistance to chemotherapy. These observations underscore the pivotal role of *WBP5* in HNSCC progression and its potential as a biomarker and therapeutic target.

Previous studies have extensively investigated *WBP5*, highlighting its involvement in various cellular and pathological processes and its critical role in several diseases. In patients with microsatellite-unstable colorectal cancer, Sanger sequencing identified mutations in the *WBP5* gene, including Arg46Lys and Glu47Gly, both located on the X chromosome [5]. However, the functional implications of these mutations remain unknown, warranting further investigation to elucidate their potential role in tumorigenesis. In macrophages of patients with coronary artery disease, *WBP5* has been identified as a key regulator of low-density lipoprotein uptake [6]. In small-cell lung cancer (SCLC), *WBP5* induces multidrug resistance (MDR) by regulating the Hippo signaling pathway, thereby activating MDR-related gene expression [4]. Targeting *WBP5* in this context may potentially mitigate MDR. In AML, overexpression of WBP5 is associated with poor survival outcomes and linked to the upregulation of HOX gene clusters, contributing to disease progression [3]. Additionally, in gastric cancer, WBP5 is a gene associated with lymph node metastasis in the advanced stages of the disease, suggesting its role as a biomarker of aggressive cancer characteristics [7].

*WBP5* overexpression is not solely associated with poor prognosis. In papillary thyroid carcinoma (PTC), higher *WBP5* expression levels have been associated with less aggressive tumor characteristics, including unilateral tumorigenesis, absence of capsule invasion, and reduced lymph node metastasis rates [8]. Patients with elevated *WBP5* expression exhibited significantly lower risks of disease recurrence compared to those with low expression levels. These contrasting findings highlight the complexity of *WBP5*’s role in cancer biology and suggest that its function may be highly tumor-specific. The dual nature of *WBP5* could be attributed to differences in tumor microenvironments, interacting signaling pathways, or cancer-specific molecular contexts. Further studies are required to elucidate the mechanisms underlying these observations and to better understand *WBP5*’s context-dependent functions in different cancer types.

Our study on HNSCC highlights the oncogenic role of *WBP5* overexpression. Significant overexpression of *WBP5* was observed in HNSCC, and it showed a positive association with *EGFR* [9,10], which serves as the target of Cetuximab [11,12], a key therapeutic agent for head and neck cancer treatment. Thus, WBP5 may contribute to the progression of HNSCC.

Cisplatin, alongside cetuximab, is extensively employed as a key therapeutic agent in the management of HNSCC. Cisplatin has demonstrated improved overall survival compared to radiotherapy alone, making it a cornerstone in HNSCC therapy [13,14,15]. Consistent with findings from studies on small-cell lung cancer (SCLC), where *WBP5* knockdown reduced drug resistance, our study reveals that the reduction in *WBP5* in HNSCC enhanced responsiveness to cisplatin.

Moreover, consistent with the findings in SCLC, where *WBP5* knockdown reduced drug resistance, the reduction in WBP5 in HNSCC increased the responsiveness to cisplatin. In SCLC, *WBP5*-mediated resistance was reported to be mitigated through the regulation of the Hippo pathway [4]. Recent genomic analyses of HNSCC tumors have reported mutations in key components of the Hippo pathway such as FAT1, WWTR1, and YAP1, which may play critical roles in the development and progression of these mutation [16].

In HNSCC, although the reduction in WBP5 enhanced drug responsiveness, the underlying mechanism remains unclear. Further research to elucidate this mechanism could establish *WBP5* as a novel therapeutic target for HNSCC treatment.

Investigating the interaction between *WBP5* and established oncogenic factors, such as *EGFR*, in HNSCC may lead to the development of new therapeutic approaches.

Future studies should focus on elucidating the molecular mechanisms through which WBP5 regulates HNSCC progression and its interaction with *EGFR*. Additionally, the development of novel small molecules targeting WBP5 may offer innovative therapeutic opportunities for patients with HNSCC.

In conclusion, our findings highlight the pivotal role of *WBP5* in the initiation and progression of HNSCC, positioning it as a key biomarker and therapeutic target for regulating cancer cell proliferation and immune evasion. Therapeutic strategies aimed at inhibiting WBP5 may be particularly effective in combination with existing anticancer agents, offering a promising breakthrough in the treatment of HNSCC and other cancers.

## 4. Materials and Methods

### 4.1. GEPIA2

The GEPIA2 version 2 platform was employed for the quantitative interrogation of WBP5 expression across diverse tissue types and its association with clinical outcomes in HNSCC. GEPIA2 (http://gepia2.cancer-pku.cn/ (accessed on 10 October 2024), an advanced web-based analytical resource, integrates datasets from The Cancer Genome Atlas (TCGA) and the Genotype-Tissue Expression (GTEx) Project, offering a robust framework for the comprehensive analysis of RNA sequencing data. This platform enables users to perform detailed analyses of gene expression patterns, compare tumor and normal tissues, and explore the relationship between gene expression and patient survival. In this study, we used GEPIA2 to analyze the differential expression of *WBP5* in HNSCC compared to that of normal tissues, and to assess the correlation between WBP5 expression levels and OS in patients with HNSCC. The tool’s capabilities for statistical analyses and interactive data visualization facilitated a robust investigation into the potential clinical relevance of WBP5 expression, as indicated in references [17,18].

### 4.2. Microarray Data

The National Center for Biotechnology Information GEO database (http://www.ncbi.nlm.nih.gov/geo/ (accessed on 28 October 2024) was used as a source of high-throughput functional genomic data. GEO is a publicly accessible resource that provides a vast collection of gene expression datasets generated using various experimental platforms and conditions. To identify relevant datasets for HNSCC, we performed a search using the keywords “head and neck squamous cell carcinoma” and “*Homo sapiens*”. From the search results, we selected several datasets that met the inclusion criteria for a comprehensive analysis. The GSE58911, GSE178537, and GSE33232 datasets were used for this study (Table 1). These datasets provided a diverse range of gene expression data that were further analyzed to assess the expression levels of WBP5 and their potential association with clinical outcomes in HNSCC [19].

### 4.3. Kaplan–Meier Plotter Analysis

A Kaplan–Meier plotter (http://kmplot.com/analysis/ (accessed on 25 October 2024) was used to evaluate the relationship between *WBP5* expression and patient survival outcomes. This bioinformatics tool integrates survival data from over 35,000 samples across 21 different cancer types, allowing for a comprehensive analysis of the prognostic value of gene expression at the mRNA, miRNA, protein, and DNA levels. In this investigation, a Kaplan–Meier plotter was employed to evaluated the relationship between *WBP5* expression levels and clinical outcomes in HNSCC patients. Survival curves were generated, and HRs with 95% confidence intervals (CI) were calculated to assess the significance of WBP5 expression on overall patient survival [23].

### 4.4. UALCAN

UALCAN (http://ualcan.path.uab.edu/ (accessed on 20 October 2024) was used as a web-based platform to analyze cancer-omics data, with a specific focus on assessing tumor gene expression levels and conducting survival analyses. This publicly accessible resource integrates data from TCGA and offers user-friendly tools for exploring gene expression across different cancer types and patient subgroups. In this study, UALCAN was used to evaluate the expression levels of *WBP5* in HNSCC samples and correlate these levels with clinical parameters, including tumor stage, grade, and patient survival [24,25].

### 4.5. TNM Plot

The TNM plot (http://www.tnmplot.com/ (accessed on 25 October 2024) was used for differential gene expression analysis of *WBP5* in HNSCC, comparing tumor tissues to normal and metastatic tissues, to understand its potential role in cancer progression and metastasis. The TNM plot is an online platform that facilitates the comparative analysis of gene expressions across tumor, normal, and metastatic tissues. It integrates data from multiple high-quality public databases, including the GEO, GTEx project, TCGA, and Therapeutically Applicable Research to Generate Effective Treatment (TARGET) initiative. The TNM plot utilized data from 56,938 unique samples comprising 15,648 normal tissue samples, 40,442 tumor samples, and 848 metastatic samples. This extensive dataset allowed for a robust and reliable evaluation of gene expression differences across various sample types [26].

### 4.6. Transfection for Small Interfering RNA (siRNA)

FaDu cells underwent transfection with WBP5-specific siRNA or a negative control (NC) siRNA procured from Bioneer (Daejeon, Republic of Korea), utilizing the RNAiMAX transfection reagent (Thermo Fisher Scientific, Waltham, MA, USA) in strict adherence to the manufacturer’s prescribed protocol. Briefly, FaDu cells were seeded, allowing us to reach the appropriate confluence, and were transfected with RNA–lipid complexes formed by mixing siRNA and RNAiMAX. The cells were then incubated under standard conditions prior to subsequent analysis. The siRNA sequences are listed in Table 2.

### 4.7. Total RNA Extraction and qRT-PCR

Isolation of total RNA was carried out using Nucleozol (Macherey-Nagel GmbH & Co., KG, Düren, Germany) in full compliance with the manufacturer’s protocol. Subsequently, complementary DNA (cDNA) synthesis was performed utilizing the PrimeScript™ RT reagent Kit (Takara, Kusatsu, Shiga, Japan) following the manufacture’s specification. Quantitative real-time PCR (qRT-PCR) was executed on a StepOnePlus real-time PCR system (Applied Biosystems, Waltham, MA, USA) employing Fast SYBR Green Master Mix (Applied Biosystems) as the detection chemistry. The relative expression levels of mRNA were calibrated to *β-actin*, used as the internal normalization control. Details of the primer sequences used in this investigation are enumerated in Table 3.

### 4.8. Cell Culturing and Viability Assessment

FaDu cells, procured from the American Type Culture Collection (ATCC), were cultured in Minimum Essential Medium (MEM) supplemented with 10% fetal bovine serum, 1% penicillin–streptomycin, and 1% sodium pyruvate. The cells were maintained at 37 °C in a humidified incubator with 5% CO_2_. Cell viability and proliferation were evaluated using the EZ-Cytox Cell Viability Assay Kit (Dogenbio, Seoul, Republic of Korea). Assays were conducted using WBP5-knockdown FaDu cells. For the cell viability experiments, 3 × 10^3^ cells/well were seeded into 96-well plates, while 1 × 10^3^ cells/well were plated for proliferation analysis. Cisplatin (Sigma-Aldrich, St. Louis, MO, USA) was introduced into each well, and the cells were incubated for 72 h. After the incubation period, 10 μL of EZ-Cytox reagent was added to each well, and absorbance was measured at 450 nm using a microplate reader (Bio-Tek, Winooski, VT, USA) [27].

### 4.9. Evaluation of Caspase 3/7 Activity

Caspase 3/7 activity was quantified utilizing the Caspase-Glo 3/7 assay kit (Promega, Mannheim, Germany). FaDu cells were seeded in white-walled 96-well plates at a density of 3 × 10^3^ cells/well and were allowed to adhere under optimal conditions. Upon sufficient cell attachment, 100 µL of Caspase-Glo 3/7 reagent was carefully introduced to each well, followed by gentle mixing to ensure uniform distribution. The reaction mixture was then incubated for 30 minutes to facilitate luminescent signal generation, which directly correlates with caspase 3/7 activity. Luminescence intensity was subsequently measured using a microplate reader (BioTek) [28,29]. 

### 4.10. Colony Formation Analysis

The colony formation potential of WBP5-knockdown FaDu cells was assessed. In summary, FaDu cells with suppressed WBP5 expression were plated into 6-well plates at a density 3 × 10^2^ cells/well. Following 48 h incubation period, cisplatin-containing culture medium was added to each well. The cells were tthen incubated for 14 days to allow for the development of individual colonies. The colonies were fixed with methanol and subsequently stained with 0.1% crystal violet solution (Sigma-Aldrich). The number of clearly visible colonies was determined using a microscope to evaluated clonogenic survival [30].

### 4.11. Statistical Analysis

Data were subjected to rigorous statistical evaluation utilizing GraphPad Prism software (version 5.0; GraphPad Software, San Diego, CA, USA). The analytical framework encompassed analysis of variance (ANOVA) for assessing group differences, followed by Tukey’s post hoc test for multiple comparisons and Student’s t-test for pairwise analysis. Results were represented as means ± standard deviations, where applicable. Statistical significance was rigorously defined at a threshold of *p* < 0.05.

## 5. Conclusions

This study underscores the critical role of *WBP5* in the progression and treatment resistance of HNSCC. Through the comprehensive analysis of publicly available datasets and experimental validation, we demonstrate that *WBP5* is significantly overexpressed in HNSCC, correlating with poor prognosis, advanced tumor stages, and metastasis. Functional studies reveal that *WBP5* promotes tumor cell proliferation, enhances chemoresistance, and modulates the tumor immune microenvironment. Notably, the strong correlation between *WBP5* and *epidermal growth factor receptor (EGFR)* expression highlights its potential as a therapeutic target. Silencing *WBP5* not only reduced tumor cell viability, but also increased cisplatin sensitivity, offering promising implications for improving treatment efficacy. Future studies should focus on elucidating the molecular mechanisms through which *WBP5* drives tumorigenesis in HNSCC and its interactions with key oncogenic pathways. Additionally, the development of therapeutic strategies targeting *WBP5* could pave the way for novel combination treatments, ultimately improving clinical outcomes for patients with HNSCC.

## Figures and Tables

**Figure 1 cancers-17-00587-f001:**
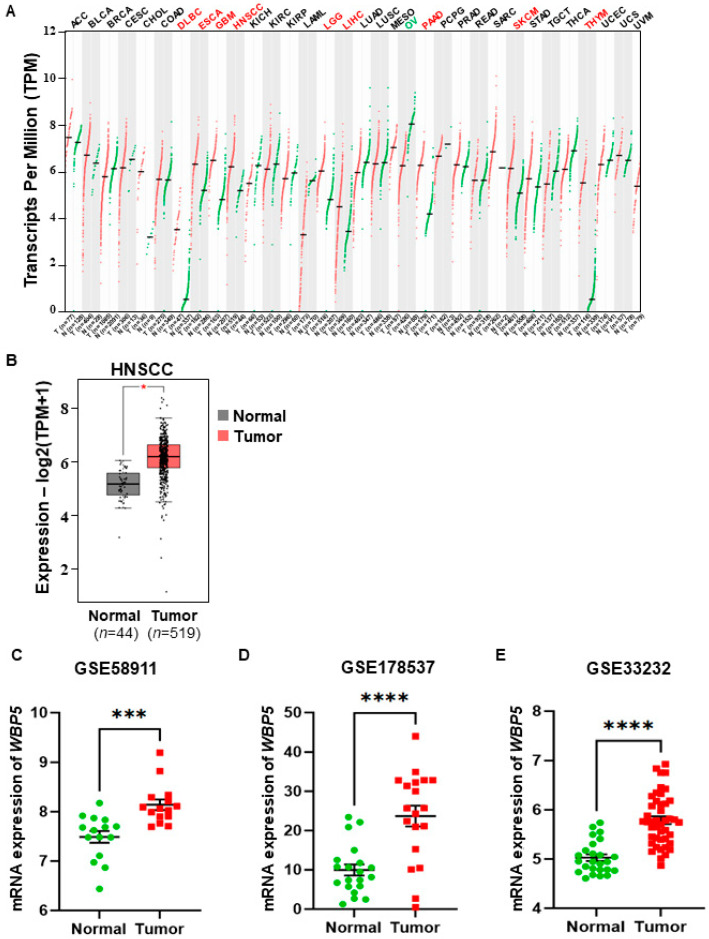
Analysis of *WBP5* expression in head and neck squamous cell carcinoma (HNSCC) using Gene Expression Profiling Interactive Analysis (GEPIA2). (**A**) *WBP5* expression in various cancers. (**B**) *WBP5* expression in HNSCC. *WBP5* expression was conducted in 44 normal samples and 519 tumor samples. The gray box denotes normal samples, while the red box indicates tumor samples. * *p* < 0.05 (**C**,**D**) The mRNA expression levels of *WBP5* were analyzed in normal and tumor tissues across multiple datasets. In each dataset, *WBP5* levels were compared between normal samples (green) and tumor samples (red): (**C**) GSE58911, (**D**) GSE178537, and (**E**) GSE33232. A significant increase in *WBP5* expression was observed in tumor samples across all datasets. Statistical significance is assessed using the appropriate test, with *p*-values represented as follows: **** *p* < 0.0001, *** *p* < 0.001.

**Figure 2 cancers-17-00587-f002:**
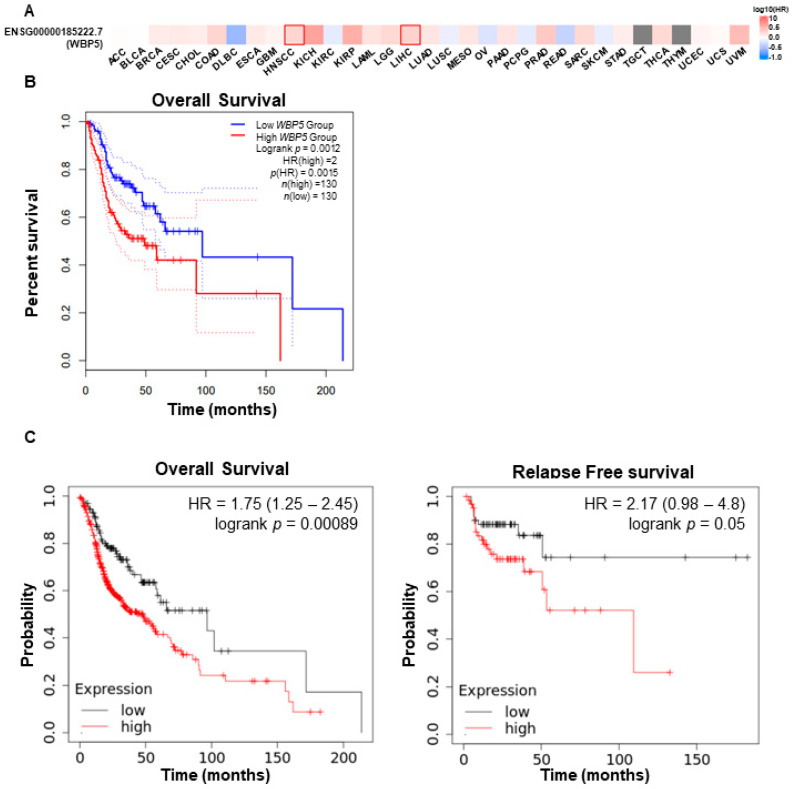
Expression and prognostic significance of *WBP5* in HNSCC. (**A**) *WBP5* expression across various cancers. Heatmap depicting *WBP5* expression levels, where high expression is indicated in red and low expression in blue. (**B**) Kaplan–Meier plot for overall survival in a specific cancer cohort, comparing high (red) and low (blue) *WBP5* expression groups. Elevated *WBP5* expression is strongly correlated with reduced survival rates (log-rank *p* = 0.0012, HR = 2.0). (**C**) Kaplan–Meier survival graphs displaying overall survival (left) and relapse-free survival (right) in cancer patients, stratified by high (red) and low (black) *WBP5* expression levels. High expression is associated with worse overall survival (HR = 1.75, log-rank *p* = 0.00089) and exhibits a trend toward worse relapse-free survival (HR = 2.17, log-rank *p* = 0.05).

**Figure 3 cancers-17-00587-f003:**
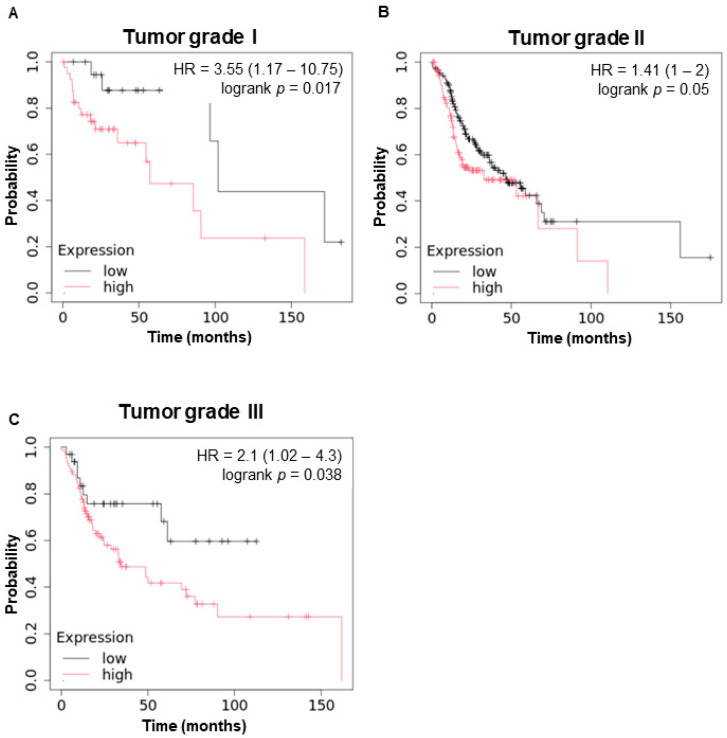
Kaplan–Meier survival curves depicting the relationship between *WBP5* expression and overall survival in patients with HNSCC across different tumor grades. (**A**) Tumor grade I: patients with low *WBP5* expression demonstrated significantly better overall survival compared to those with high expression (HR = 3.55, 95% CI: 1.17–10.75, log-rank *p* = 0.017). The number of patients at risk at various time points is indicated at the bottom of the plot. (**B**) Tumor grade II: high *WBP5* expression was associated with poorer overall survival in patients with grade II head and neck cancer (HR = 1.41, 95% CI: 1.0–2), with a log-rank *p* of 0.05. The number of patients at risk is presented at the bottom. (**C**) Tumor grade III: Kaplan–Meier analysis for patients with grade III head and neck cancer indicated that those with high *WBP5* expression had worse survival outcomes (HR = 2.1, 95% CI: 1.02–4.3, log-rank *p* = 0.038). The number of patients at risk over time is presented below the curve.

**Figure 4 cancers-17-00587-f004:**
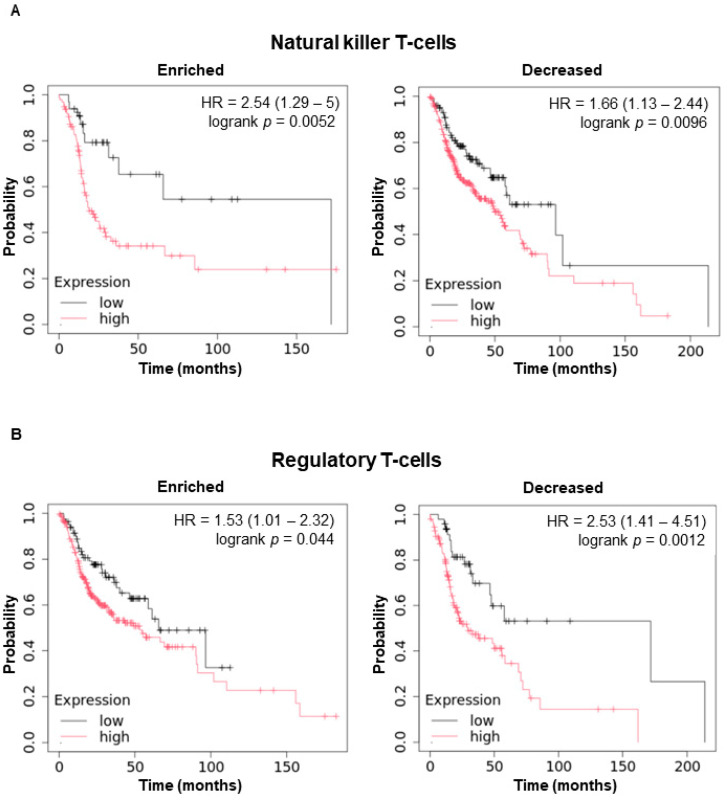
Survival plots based on the Kaplan–Meier analysis illustrating the effect of high versus low expressions of immune cell subsets on overall survival in HNSCC patients. (**A**) A high expression of NK T-cells was linked to poorer overall survival in both enriched (HR = 2.54, *p* = 0.0052) and decreased groups (HR = 1.66, log-rank *p* = 0.0096). (**B**) High Treg expression negatively impacted survival in both the enriched (HR = 1.53, log-rank *p* = 0.044) and decreased groups (HR = 2.53, log-rank *p* = 0.0012). (**C**) High CD4+ memory T-cell expression correlated with worse survival in enriched (HR = 1.86, log-rank *p* = 0.0048) and decreased groups (HR = 2.05, log-rank *p* = 0.0025). (**D**) High macrophage expression was associated with reduced survival in the enriched (HR = 1.73, log-rank *p* = 0.01) and decreased groups (HR = 1.93, log-rank *p* = 0.0033).

**Figure 5 cancers-17-00587-f005:**
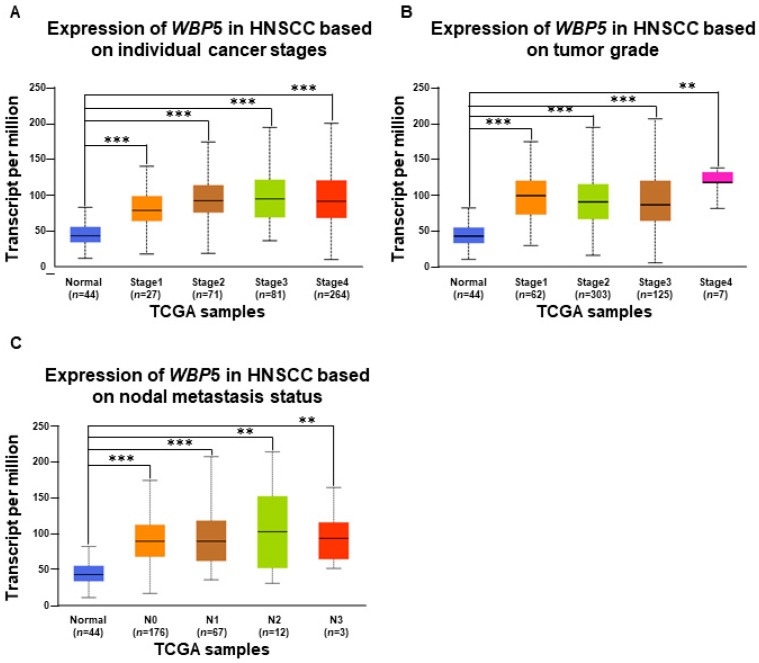
Analysis of *WBP5* expression in HNSCC across cancer stages, tumor grade, and nodal metastasis status. (**A**) Box plot depicting *WBP5* expression levels across HNSCC stages (from 1 to 4) versus normal tissue, based on TCGA samples. Expression significantly increases with cancer stage, peaking at Stage 4 (*** *p* < 0.001, ** *p* < 0.01). (**B**) Box plot of *WBP5* expression across tumor grades (from 1 to 4) compared to normal tissue, exhibiting higher expression in higher-grade tumors, especially Grade 4 (*** *p* < 0.001, ** *p* < 0.01). (**C**) Box plot illustrating *WBP5* expression by nodal metastasis status (N0 to N3). Significant expression increases in N1 and N2, with a decrease in N3 (*** *p* < 0.001 and ** *p* < 0.01).

**Figure 6 cancers-17-00587-f006:**
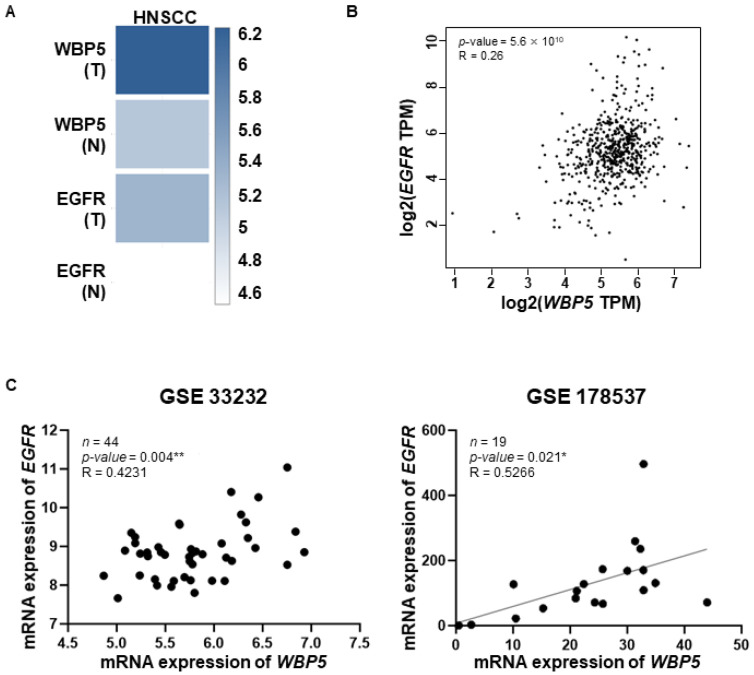
Correlation of *WBP5* and *EGFR* expression across different cancer types and datasets. (**A**) Heatmaps of *WBP5* and *EGFR* expression across HNSCC, with darker blue indicating higher expression levels. (**B**) Scatter plot showing the correlation between *WBP5* and *EGFR* expression (log2 TPM) across samples, with a positive correlation (R = 0.26, *p* = 5.6 × 10^10^). (**C**) Scatter plots showing the correlation between *WBP5* and *EGFR* expression in specific datasets (GSE33232 and GSE178537). Both datasets demonstrate a positive correlation between *WBP5* and *EGFR* expression, with GSE33232 (R = 0.4231, *p* = 0.004) and GSE178537 (R = 0.5266, *p* = 0.021) (* *p* < 0.05 and ** *p* < 0.01).

**Figure 7 cancers-17-00587-f007:**
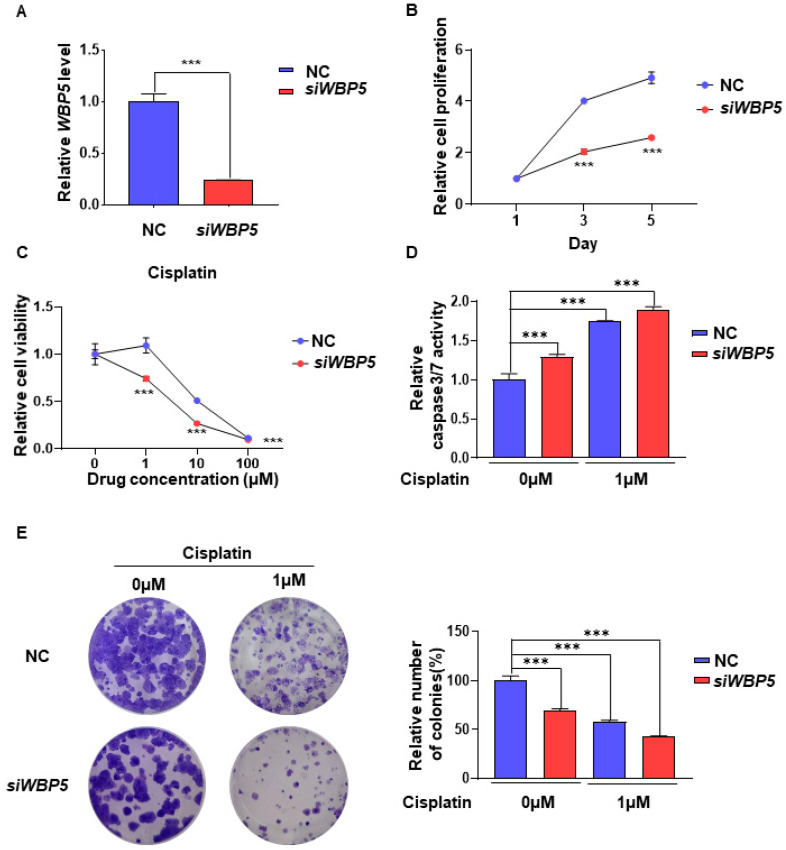
Knockdown of *WBP5* reduces cell proliferation and enhances sensitivity to cisplatin in HNSCC cells. (**A**) *WBP5* mRNA levels are significantly downregulated in *siWBP5*-transfected cells compared to NC (*** *p* < 0.001). (**B**) Cell proliferation is significantly reduced in *siWBP5* cells over 1, 3, and 5 days compared to NC (*** *p* < 0.001). (**C**) Cell viability decreases with increasing cisplatin concentrations, with a greater reduction in *siWBP5* cells (*** *p* < 0.001). (**D**) Caspase 3/7 activity is higher in *siWBP5* cells, especially after 1 μM cisplatin treatment, indicating increased apoptosis (*** *p* < 0.001). (**E**) Clonogenic assay results showing colony formation in NC- and *siWBP5*-treated cells with 0µM and 1µM cisplatin. *siWBP5*-treated cells demonstrate reduced colony formation, particularly with cisplatin exposure. Quantification is presented on the right (*** *p* < 0.001).

**Table 1 cancers-17-00587-t001:** Gene Expression Omnibus (GEO) dataset for head and neck squamous cell carcinoma in *Homo sapiens*.

Reference	GEO	Platform	Sample	Normal	Tumor	Metastasis
Lobert et al. [20]	GSE58911	GPL6244	Head and neck squamous cell carcinoma	15	15	
Cheng et al. [21]	GSE178537	GPL16791 GPL18573	Head and neck squamous cell carcinoma	20	19	9
Stansfield et al. [22]	GSE33232	GPL5175	Head and neck squamous cell carcinoma	25	44	

**Table 2 cancers-17-00587-t002:** Summary of siRNA sequences utilized in this study.

Gene	Sense	Antisense
*s* *iWBP5*	CUGGUCAUCUGGUCCUUGU	ACAAGGACCAGAUUACCAG

**Table 3 cancers-17-00587-t003:** Summary of quantitative reverse transcription polymerase chain reaction primers utilized in this study.

Gene	Sense	Antisense
*WBP5*	AGCTAGAGGAGGAGGCCAAA	TCCTGGAGAGATTGAATCAGCC
*β-actin*	TCCTCTCCCAAGTCCACACAGG	GGGCACGAAGGCTCATCATTC

## Data Availability

The data that support the findings of this study are available from the corresponding author upon request.

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
