# Peer review of "WBP5 Expression Influences Prognosis and Treatment Response in Head and Neck Squamous Cell Carcinoma"

_cancers, 2025, doi:10.3390/cancers17040587_

Round 1
Reviewer 1 Report
Comments and Suggestions for Authors
The topic, objectives, and results are clear.
It is suitable for the audience and the journal.
The article follows a logical and well-defined structure. These findings suggest that WBP5 may serve as a prognostic biomarker and potential therapeutic target in HNSCC. Modulating WBP5 expression may represent a novel strategy to enhance treatment efficacy. Future studies should elucidate the precise mechanisms of WBP5 action and develop targeted therapies. This integrated approach, combining comprehensive analysis of publicly available datasets with in vitro experimental validation provides strong evidence for the clinical significance of WBP5 in HNSCC
The study likely has novel elements
- WBP5 has not been previously explored in depth in HNSCC.
- The relationship between WBP5, chemoresistance, and EGFR has not been reported before.
- The proposed therapeutic strategies are innovative and have not been extensively investigated.
The methodology is clearly and comprehensibly described.
The study design is appropriate to address the objectives and research questions.
The tools used are clearly identified.
Why were GEPIA2, UALCAN, and the GEO datasets chosen for this analysis?
What are the main limitations of this study that could affect its clinical applicability?
What specific steps are suggested to develop therapies targeting WBP5?
Author Response
Dear Reviewer,
We would like to express our deepest gratitude for taking the time to review our manuscript, "The Impact of WBP5 Expression on the Prognosis and Therapeutic Response of Head and Neck Squamous Cell Carcinoma."
We greatly appreciate your thoughtful and constructive comments, which have been invaluable in enhancing the quality of our work. We have made every effort to address all the reviewers' concerns with the utmost care and diligence.
Your prompt feedback and valuable advice on improving and refining the manuscript have been immensely helpful, and we are sincerely grateful for your guidance throughout the process.
We hope that the revised version of our manuscript will meet the high standards of Cancer and be deemed suitable for publication.
Thank you once again for your time, expertise, and dedication to the review process.
Sincerely,
Yeon Soo Kim, M.D., Ph.D.
Department of Otorhinolaryngology–Head and Neck Surgery, Korea University College of Medicine, Seoul, Korea
E-mail: ionskim@korea.ac.kr
# Reviewer 1.
The topic, objectives, and results are clear. It is suitable for the audience and the journal.
The article follows a logical and well-defined structure. These findings suggest that WBP5 may serve as a prognostic biomarker and potential therapeutic target in HNSCC. Modulating WBP5 expression may represent a novel strategy to enhance treatment efficacy. Future studies should elucidate the precise mechanisms of WBP5 action and develop targeted therapies. This integrated approach, combining comprehensive analysis of publicly available datasets with in vitro experimental validation provides strong evidence for the clinical significance of WBP5 in HNSCC
The study likely has novel elements WBP5 has not been previously explored in depth in HNSCC. The relationship between WBP5, chemoresistance, and EGFR has not been reported before. The proposed therapeutic strategies are innovative and have not been extensively investigated. The methodology is clearly and comprehensibly described.
The study design is appropriate to address the objectives and research questions.The tools used are clearly identified.
∙ Why were GEPIA2, UALCAN, and the GEO datasets chosen for this analysis?
Response:
GEPIA2, UALCAN, and the GEO datasets were chosen for this analysis because each platform provides unique strengths that complement one another, enabling a comprehensive and reliable approach.
- GEPIA2 integrates TCGA and GTEx data, allowing easy comparison of gene expression levels between tumor and normal tissues. It is user-friendly, accessible even to non-experts, and supports advanced bioinformatics analyses such as survival analysis and gene correlation studies.
- UALCAN was selected for its capability to perform subgroup-specific analyses within TCGA data, enabling more stratified and detailed insights into tumor biology.
- GEO datasets were utilized because they provide a large repository of global microarray and RNA-seq data from diverse diseases, tissues, and conditions. These datasets offer the flexibility to analyze existing data, reducing research costs and time while enabling custom analyses tailored to the study's needs.
By combining these platforms, we leveraged their individual advantages to enhance the reliability of our results through cross-validation. This approach also allowed us to perform diverse analyses, including survival analysis, expression comparisons, and correlation studies, ultimately accelerating hypothesis validation and adding depth to our findings.
∙ What are the main limitations of this study that could affect its clinical applicability?
Response:
While our study demonstrated that WBP5 expression could influence the prognosis and treatment outcomes of head and neck squamous cell carcinoma, there are several limitations that could affect its clinical applicability. First, the underlying mechanisms by which WBP5 exerts its effects remain unclear, which limits its immediate translation into clinical strategies. Second, the findings are primarily based on TCGA and GTEx data, and the reproducibility of these results in broader and more diverse patient cohorts has not yet been confirmed. Lastly, the in vitro experiments provide important insights but may not fully replicate the complexity of in vivo conditions. Addressing these limitations in future studies will be crucial to fully establish WBP5 as a viable therapeutic target in clinical settings.
∙ What specific steps are suggested to develop therapies targeting WBP5?
Response:
First, it is essential to elucidate the molecular mechanisms of WBP5 and validate its effects in in vivo models. This will provide a deeper understanding of how WBP5 influences tumor progression and treatment outcomes. By identifying the signaling pathways involving WBP5 and confirming its effects through in vivo experiments, the potential of WBP5 as a therapeutic target will be significantly enhanced. Furthermore, designing small molecules or biologics that regulate WBP5, and demonstrating that their effects are consistent with the identified mechanisms and in vivo findings, could pave the way for the development of novel targeted therapies. These steps, coupled with further validation in diverse patient-derived models, will be critical for translating these findings into clinical applications.
Reviewer 2 Report
Comments and Suggestions for Authors
This manuscript entitled "WBP5 Expression Influences Prognosis and Treatment Response in Head and Neck Squamous Cell Carcinoma" submitted by Jeong discusses a thorough investigation into the role of WBP5 as a potential therapeutic target in head and neck squamous cell carcinoma (HNSCC). The study leverages bioinformatics analysis using publicly available datasets, such as Gene Expression Profiling Interactive Analysis and UALCAN, to demonstrate that WBP5 is significantly overexpressed in advanced stages and higher tumor grades of HNSCC. The authors also establish a correlation between elevated WBP5 expression and poor patient prognosis, which is further supported by functional validation in vitro. These findings are compelling, as they suggest that WBP5 could serve as both a prognostic biomarker and a novel target for therapeutic intervention, especially in overcoming chemoresistance.
The experimental results showing that silencing WBP5 in FaDu cells not only reduces cell proliferation but also enhances sensitivity to cisplatin provide a solid foundation for future research into targeting WBP5 as a strategy to improve treatment outcomes in HNSCC. under the figure 7 the authors need to show expression levels of WBP5 protein levels after the use of WBP5 siRNA. However, the manuscript would benefit from further exploration of the molecular mechanisms underlying WBP5's role in tumor progression and chemoresistance. While the study makes a strong case for the clinical relevance of WBP5, additional in vivo validation and the development of specific inhibitors would be crucial next steps. Overall, this work makes a valuable contribution to the field and opens the door for future targeted therapeutic strategies in HNSCC and is commendable.Author Response
Dear Reviewer,
We would like to express our deepest gratitude for taking the time to review our
manuscript, "The Impact of WBP5 Expression on the Prognosis and Therapeutic
Response of Head and Neck Squamous Cell Carcinoma."
We greatly appreciate your thoughtful and constructive comments, which have been
invaluable in enhancing the quality of our work. We have made every effort to address all
the reviewers' concerns with the utmost care and diligence.
Your prompt feedback and valuable advice on improving and refining the manuscript
have been immensely helpful, and we are sincerely grateful for your guidance throughout
the process.
We hope that the revised version of our manuscript will meet the high standards of
Cancer and be deemed suitable for publication.
Thank you once again for your time, expertise, and dedication to the review process.
Sincerely,
Yeon Soo Kim, M.D., Ph.D.
Department of Otorhinolaryngology–Head and Neck Surgery, Korea University College of
Medicine, Seoul, Korea
E-mail: ionskim@korea.ac.kr
# Reviewer 2.
This manuscript entitled "WBP5 Expression Influences Prognosis and Treatment Response
in Head and Neck Squamous Cell Carcinoma" submitted by Jeong discusses a thorough
investigation into the role of WBP5 as a potential therapeutic target in head and neck
squamous cell carcinoma (HNSCC). The study leverages bioinformatics analysis using
publicly available datasets, such as Gene Expression Profiling Interactive Analysis and
UALCAN, to demonstrate that WBP5 is significantly overexpressed in advanced stages and
higher tumor grades of HNSCC. The authors also establish a correlation between elevated
WBP5 expression and poor patient prognosis, which is further supported by functional
validation in vitro. These findings are compelling, as they suggest that WBP5 could serve as
both a prognostic biomarker and a novel target for therapeutic intervention, especially in
overcoming chemoresistance.
The experimental results showing that silencing WBP5 in FaDu cells not only reduces cell
proliferation but also enhances sensitivity to cisplatin provide a solid foundation for future
research into targeting WBP5 as a strategy to improve treatment outcomes in HNSCC. under
the figure 7 the authors need to show expression levels of WBP5 protein levels after the use of
WBP5 siRNA. However, the manuscript would benefit from further exploration of the
molecular mechanisms underlying WBP5's role in tumor progression and chemoresistance.
While the study makes a strong case for the clinical relevance of WBP5, additional in vivo
validation and the development of specific inhibitors would be crucial next steps. Overall,
this work makes a valuable contribution to the field and opens the door for future targeted
therapeutic strategies in HNSCC and is commendable.
Response:
We sincerely thank the reviewer for the thoughtful and constructive feedback on our
manuscript. Our study aimed to explore the potential of WBP5 as a prognostic biomarker and
therapeutic target in head and neck squamous cell carcinoma (HNSCC) using publicly
available datasets.
- Comment on WBP5 protein levels after siWBP5 treatment:
We appreciate the reviewer’s suggestion regarding the necessity of presenting WBP5 protein
expression levels following siRNA treatment. In this study, we focused on mRNA-level
comparisons to support WBP5’s potential as a therapeutic target. We acknowledge that
including protein-level data would further strengthen the findings, and we plan to incorporate
this aspect into our future research.
- Comment on exploring the molecular mechanisms of WBP5:
We fully agree with the reviewer that elucidating the molecular mechanisms of WBP5 is
critical for understanding its role in tumor progression and chemoresistance. While our
current study focused on establishing the initial functional validation of WBP5, we recognize
the importance of further exploring its downstream signaling pathways. In future studies, we
plan to conduct in vivo experiments to validate the biological relevance of WBP5 in tumor
progression and treatment response, including its role in chemoresistance.
Additionally, we aim to develop specific small molecule inhibitors or biologics that target
WBP5, which could provide a foundation for novel therapeutic strategies. By combining
these approaches—mechanistic studies, in vivo validation, and inhibitor development—we
believe we can further strengthen the clinical relevance and therapeutic potential of WBP5 in
head and neck squamous cell carcinoma.
We are grateful for the reviewer’s positive remarks regarding the value of our study and its
contribution to advancing targeted therapeutic strategies in HNSCC. Your feedback has
provided valuable insights that will guide the next steps of our research.
Reviewer 3 Report
Comments and Suggestions for Authors
Comments and Suggestions for Authors
Authors reported in clinical study analysis relationship between WBP5 expression and prognosis and treatment in HNSCC. I appreciate the novelty and topic of research chosen by the Authors. Authors are very smoothly describing each part of the manuscript.
Introduction:
Lines 44-51: Could you please provide more information about head and neck cancers?
Please provide an abbreviation for HPV on lines 47-48.
Lines 60-62: In the sentence The Authors write "...driver along 14 other genes, including...", shouldn't the gene names be italicized?
Lines 65-66: Please provide the gene names in italics.
The entire introduction lacks appropriate references based on which the sentences were written.
Results:
In whole manuscript, “p” in p-values should be written in italics.
Line 113: Is the p before “(hazard ratio ....)” necessary?
Discussion:
The discussion is well written and shows the future perspective and conclusions. Were there any limitations to this study?
Author Response
Dear Reviewer,
We would like to express our deepest gratitude for taking the time to review our manuscript, "The Impact of WBP5 Expression on the Prognosis and Therapeutic Response of Head and Neck Squamous Cell Carcinoma."
We greatly appreciate your thoughtful and constructive comments, which have been invaluable in enhancing the quality of our work. We have made every effort to address all the reviewers' concerns with the utmost care and diligence.
Your prompt feedback and valuable advice on improving and refining the manuscript have been immensely helpful, and we are sincerely grateful for your guidance throughout the process.
We hope that the revised version of our manuscript will meet the high standards of Cancer and be deemed suitable for publication.
Thank you once again for your time, expertise, and dedication to the review process.
Sincerely,
Yeon Soo Kim, M.D., Ph.D.
Department of Otorhinolaryngology–Head and Neck Surgery, Korea University College of Medicine, Seoul, Korea
E-mail: ionskim@korea.ac.kr
# Reviewer 3.
Authors reported in clinical study analysis relationship between WBP5 expression and prognosis and treatment in HNSCC. I appreciate the novelty and topic of research chosen by the Authors. Authors are very smoothly describing each part of the manuscript.
∙ Introduction:
∙ Lines 44-51: Could you please provide more information about head and neck cancers?
Response:
Thank you for your thoughtful feedback on our manuscript. In response to the reviewer's request, we have revised and expanded the section on head and neck squamous cell carcinoma (HNSCC) to provide a more detailed and comprehensive overview. The updated section now includes additional information on the molecular mechanisms underlying HNSCC pathogenesis, such as genetic and epigenetic alterations (e.g., mutations in TP53 and CDKN2A, as well as aberrant signaling in pathways like EGFR, PI3K/AKT, and NOTCH). We have also emphasized the ongoing therapeutic challenges, such as intrinsic and acquired resistance, and underscored the critical need for more effective treatment strategies due to the persistently low 5-year survival rate of approximately 50%."
We believe that these revisions provide a more robust foundation for our study and further emphasize its significance within the context of HNSCC research. Thank you again for your valuable suggestion.
∙ Please provide an abbreviation for HPV on lines 47-48.
Response:
Thank you for pointing out the missing aspects in the manuscript. We have revised the sentences from the previous question accordingly and made the necessary adjustments. We appreciate your valuable input.
Before; Page 2, line47 - 48
The primary causes of HNSCC are strongly associated with smoking, alcohol consumption, and human papillomavirus infection
After ; Page 2, lines 49 -51
The etiology of HNSCC is strongly linked to tobacco use, chronic alcohol consumption, and human papillomavirus (HPV) infection.
∙ Lines 60-62: In the sentence The Authors write "...driver along 14 other genes, including...", shouldn't the gene names be italicized?
Response:
Before; Page 2, line 60 – 42
Sanger sequencing has identified mutations in WBP5 on the X chromosome in pa60 tients with colorectal cancer with microsatellite instability, implicating it as a potential 61 mutational driver along with 14 other genes, including ADAR, DCAF12L2, GLT1D1, 62 ITGA7, MAP1B, MRGPRX4, PSRC1, RANBP2, RPS6KL1, SNCAIP, TCEAL6, TUB6, 63 VEGFB, and ZBTB2 [2].
After ; Page 2, lines 64 - 66
Sanger sequencing has identified mutations in WBP5 on the X chromosome in patients with colorectal cancer with microsatellite instability, implicating it as a potential mutational driver along with 14 other genes, including ADAR, DCAF12L2, GLT1D1, ITGA7, MAP1B, MRGPRX4, PSRC1, RANBP2, RPS6KL1, SNCAIP, TCEAL6, TUB6, VEGFB, and ZBTB2 [4, 5].
Added reference
- Tang R, Lei Y, Hu B, Yang J, Fang S, Wang Q, Li M, Guo L: WW domain binding protein 5 induces multidrug resistance of small cell lung cancer under the regulation of miR-335 through the Hippo pathway. Br J Cancer 2016, 115(2):243-251.
∙ Lines 65-66: Please provide the gene names in italics.
The entire introduction lacks appropriate references based on which the sentences were written.
Response:
Thank you for identifying the gaps in the manuscript. We have updated the sentences in response to the previous question and made the required revisions.
Before; Page 2, line 65 – 66
Additionally, in acute myeloid leukemia (AML), WBP5 overexpression has been associated with poor prognosis, correlating with increased expression of HOX gene cluster (HOXA5, HOXA7, HOXA9, and HOXA10), MEIS1, and FOXC1 [4].
After ; Page 2, lines 69 -72
Additionally, in acute myeloid leukemia (AML), WBP5 overexpression has been associated with poor prognosis, correlating with increased expression of HOX gene cluster (HOXA5, HOXA7, HOXA9, and HOXA10), MEIS1, and FOXC1 [3].
∙ Results:
∙ In whole manuscript, “p” in p-values should be written in italics.
Response:
Thank you for your comment. We have carefully reviewed the manuscript and updated all instances of “p” in p-values to italics as requested. We kindly ask for your understanding as we did not mark the revised sections separately due to the extensive nature of the changes. We appreciate your attention to detail and the opportunity to improve the consistency of our work."
∙ Line 113: Is the p before “(hazard ratio ....)” necessary?
Response:
Thank you for pointing this out. We have reviewed and corrected the text to ensure consistency and clarity. The unnecessary 'p' before “(hazard ratio ...)” has been removed in the revised manuscript.
Before; Page 4, line 113
The overall survival (OS) graph also showed that WBP5 overexpression was associated with reduced patent survival (p(hazard ratio [HR])=0.0015, p = 0.0012) (Figure 2B).
After ; Page 4, lines 117
The overall survival (OS) graph also showed that WBP5 overexpression was associated with reduced patent survival (hazard ratio [HR] = 0.0015, p = 0.0012) (Figure 2B).
∙ Discussion:
The discussion is well written and shows the future perspective and conclusions. Were there any limitations to this study?
Response:
Thank you for your kind feedback on the discussion section. While our study provides valuable insights into the role of WBP5 in head and neck squamous cell carcinoma, there are some limitations to acknowledge. First, the molecular mechanisms by which WBP5 influences tumor progression and chemoresistance remain unclear and require further investigation. Second, although we validated our findings through bioinformatics analyses and in vitro experiments, the lack of in vivo validation limits the immediate translational applicability of our results. We plan to overcome these limitations in our future research.
Reviewer 4 Report
Comments and Suggestions for Authors
This manuscript was entitled as “WBP5 Expression Influences Prognosis and Treatment Response in Head and Neck Squamous Cell Carcinoma” The authors concluded that WBP5 may serve as a prognostic biomarker and potential therapeutic target in HNSCC. Modulating WBP5 expression may represent a novel strategy to enhance treatment efficacy.
1. The authors gave a comprehensive evaluation of the roles of WBP5 in head and neck squamous cell carcinoma using datasets.
2. On figures 2, 3 and 4, the authors did not define high and low WBP5 expression. The authors should give a clear description of high and low WBP5 expression.
3. On figure 5, the WBP5 expression did not increase with the cancer stage, tumor grade and nodal metastasis status. It seems that WBP5 expression is not related to the cancer burden or severity. The authors should give more comprehensive explanation of the results of Figure 5.
Author Response
Dear Reviewer,
We would like to express our deepest gratitude for taking the time to review our manuscript, "The Impact of WBP5 Expression on the Prognosis and Therapeutic Response of Head and Neck Squamous Cell Carcinoma."
We greatly appreciate your thoughtful and constructive comments, which have been invaluable in enhancing the quality of our work. We have made every effort to address all the reviewers' concerns with the utmost care and diligence.
Your prompt feedback and valuable advice on improving and refining the manuscript have been immensely helpful, and we are sincerely grateful for your guidance throughout the process.
We hope that the revised version of our manuscript will meet the high standards of Cancer and be deemed suitable for publication.
Thank you once again for your time, expertise, and dedication to the review process.
Sincerely,
Yeon Soo Kim, M.D., Ph.D.
Department of Otorhinolaryngology–Head and Neck Surgery, Korea University College of Medicine, Seoul, Korea
E-mail: ionskim@korea.ac.kr
# Reviewer 4.
This manuscript was entitled as “WBP5 Expression Influences Prognosis and Treatment Response in Head and Neck Squamous Cell Carcinoma” The authors concluded that WBP5 may serve as a prognostic biomarker and potential therapeutic target in HNSCC. Modulating WBP5 expression may represent a novel strategy to enhance treatment efficacy.
- The authors gave a comprehensive evaluation of the roles of WBP5 in head and neck squamous cell carcinoma using datasets.
Response:
Thank you for accurately understanding the objectives of our study and for providing thoughtful and constructive feedback.
- On figures 2, 3 and 4, the authors did not define high and low WBP5 expression. The authors should give a clear description of high and low WBP5 expression.
Response:
Thank you for your valuable feedback. Figures 2, 3, and 4 represent analyses of overall survival (OS) based on clinical data and gene expression profiles obtained from publicly available datasets. These figures evaluate the relationship between WBP5 expression and survival outcomes in patients with HNSCC. In response to your suggestions, we have revised the Results section to include additional details about the dataset and methodology, as well as the sample size and analytical approach used for these figures. Specifically, we clarified how patients were stratified into high and low WBP5 expression groups, based on median expression levels within the dataset, and provided a more comprehensive description of the survival analysis results. These revisions aim to enhance the clarity and depth of the analysis, ensuring that the sample characteristics and results are presented with greater transparency. We appreciate your insightful suggestions, which have significantly improved the overall quality of our manuscript. Please review the updated content, and let us know if further adjustments are needed. Thank you again for your thoughtful input.
Before; Page 3, line 106
The prognostic significance of WBP5 expression in HNSCC was comprehensively evaluated using Kaplan–Meier survival analyses, which revealed a strong correlation between higher WBP5 expression and poorer clinical outcomes.
We used the GEPIA2 database and Kaplan–Meier survival analysis to assess the prognostic significance of WBP5 expression in HNSCC. Analysis of the survival map from GEPIA2 revealed that an elevated WBP5 expression level was associated with reduced survival rates in patients with HNSCC and liver hepatocellular carcinoma (LIHC) (Figure 2A). The overall survival (OS) graph also showed that WBP5 overexpression was associated with reduced patent survival (p(hazard ratio [HR])=0.0015, p = 0.0012) (Figure 2B). Notably, further Kaplan–Meier analysis yielded findings consistent with the GEPIA2 database results. Patients with higher WBP5 expression levels showed significantly reduced OS (HR = 1.75, p = 0.00089) and relapse-free survival (HR = 2.17, p = 0.05).
Stratified analyses confirmed a consistent link between elevated WBP5 expression levels and poor survival across all tumor grades. In grade I tumors, a high WBP5 expression level was linked to a substantially increased mortality risk (HR = 3.55, p = 0.017), suggesting that WBP5 may play a critical role even in early stage disease progression. Similarly, for grade II tumors, a high WBP5 expression level was significantly associated with worse survival outcomes (HR = 1.41, p = 0.05). This trend persisted in advanced grade III tumors, where high WBP5 levels were predictive of poorer survival (HR = 2.1, p = 0.038) (Figure 3). These findings demonstrate the consistent prognostic effect of WBP5 expression, showing its harmful effect on survival across different stages of tumor progression, and indicating that WBP5 contributes to poor prognosis in HNSCC. Further analyses stratified by sex revealed an association with high WBP5 expression levels. Further sex-stratified analyses confirmed that elevated WBP5 expression levels were associated with reduced survival rates, irrespective of sex (Supplementary Figure 1A, B). These results emphasize the importance of WBP5 as a prognostic indicator and its potential role in shaping personalized treatment approaches for patients with HNSCC.
After; Page 3, line 111
The prognostic relevance of WBP5 expression in head and neck squamous cell carcinoma (HNSCC) was thoroughly evaluated through Kaplan–Meier survival analyses, revealing a strong association between elevated WBP5 expression and poor clinical outcomes. Using the GEPIA2 database, we analyzed survival data for HNSCC and other cancers. The survival map generated from GEPIA2 indicated that higher WBP5 expression levels were significantly associated with reduced survival rates in patients with HNSCC and liver hepatocellular carcinoma (LIHC) (Figure 2A). Specifically, Kaplan–Meier analysis of overall survival (OS) demonstrated that elevated WBP5 expression was linked to significantly worse survival outcomes (hazard ratio [HR] = 0.0015, p = 0.0012) (Figure 2B). To validate these findings, we conducted additional Kaplan–Meier survival analyses, which were consistent with the GEPIA2 results. Patients with high WBP5 expression exhibited significantly reduced OS (HR = 1.75, 95% CI: 1.25–2.45, log-rank p = 0.00089) and relapse-free survival (RFS) (HR = 2.17, 95% CI: 0.98–4.8, log-rank p = 0.05) (Figure 2C). These findings underscore the significant role of WBP5 in negatively influencing both OS and RFS outcomes in HNSCC. Further stratified analyses examined the relationship between WBP5 expression and survival outcomes across different tumor grades. In grade I tumors, patients with high WBP5 expression had a substantially higher risk of mortality compared to those with low WBP5 expression (HR = 3.55, 95% CI: 1.17–10.75, log-rank p = 0.017) (Figure 3A). This suggests that WBP5 may have a critical role in driving early-stage tumor progression. For grade II tumors, high WBP5 expression levels were also significantly associated with reduced survival rates (HR = 1.41, 95% CI: 1–2, log-rank p = 0.05) (Figure 3B). Similarly, in advanced grade III tumors, elevated WBP5 levels were predictive of poorer survival outcomes (HR = 2.1, 95% CI: 1.02–4.3, log-rank p = 0.038) (Figure 3C). These consistent trends across tumor grades highlight WBP5’s pivotal role in promoting tumor progression and its prognostic value in HNSCC. In addition, sex-stratified analyses were performed to explore the influence of WBP5 expression on survival outcomes in male and female patients with HNSCC (Supplementary Figure 1). Among male patients, those with high WBP5 expression levels exhibited significantly worse overall survival compared to those with low WBP5 expression (HR = 1.63, 95% CI: 1.15–2.32, log-rank p = 0.0054) (Supplementary Figure 1A). Similarly, in female patients, high WBP5 expression was also associated with reduced survival (HR = 1.9, 95% CI: 1.06–3.43, log-rank p = 0.029) (Supplementary Figure 1B). These results demonstrate that elevated WBP5 expression negatively impacts survival outcomes across sexes, further supporting its role as a robust prognostic biomarker in HNSCC. Taken together, these findings strongly suggest that WBP5 expression serves as a reliable prognostic biomarker across various clinical and pathological parameters in HNSCC. The consistent association between high WBP5 expression and poor survival outcomes underscores its potential role in guiding personalized treatment strategies for HNSCC patients. Further studies are warranted to explore the underlying mechanisms by which WBP5 contributes to tumor progression and poor prognosis, and to assess its utility as a therapeutic target.
Before; Page 6, line 157
The association between WBP5 expression and immune cell infiltration was analyzed to explore its potential role in the tumor microenvironment and its effect on immune-mediated responses. Immune cell infiltration plays a critical role in tumor progression, immune evasion, and patient outcomes because infiltrating immune cells can either suppress or promote tumor growth depending on their subtype and functional state. Understanding how WBP5 expression influences or correlates with the infiltration of specific immune cell populations can elucidate its role in modulating the tumor-immune interface and its potential as a target for immunotherapy.
Elevated WBP5 expression levels were associated with significantly poorer survival outcomes in patients with both enriched and decreased levels of key immune cell subtypes including natural killer T (NKT) cells, regulatory T cells, CD4+ memory T cells, and macrophages (Figure 4). A high WBP5 expression level was associated with an increased risk of mortality in patients with enriched (HR = 2.54, p = 0.0052) and decreased NKT cells (HR = 1.66, p = 0.0096) (Figure 4A). Thus, elevated WBP5 expression levels may impair the antitumor activity of NKT cells, that play a role in the recognition and elimination of malignant cells.
A similar association was observed in regulatory T cells. In patients with enriched regulatory T cell populations, high WBP5 expression levels predicted poor survival (HR = 1.53, p = 0.044), whereas in those with decreased regulatory T cells, the HR was even higher (HR = 2.53, p = 0.0012). Thus, WBP5 may influence the immunosuppressive functions of regulatory T cells, thereby contributing to tumor immune evasion (Figure 4B).
The relationship between WBP5 expression and CD4+ memory T cells was examined. A high WBP5 expression level was associated with worse outcomes in patients with both enriched (HR = 1.86, p = 0.0048) and decreased (HR = 2.05, p = 0.0025) CD4+ memory T cell levels, highlighting its broad impact on adaptive immune responses (Figure 4C).
In cancer, macrophages exhibit dual characteristics functioning as either pro-tumorigenic (M2-like) or antitumorigenic (M1-like) entities. When WBP5 was overexpressed, patient survival was reduced regardless of the presence or absence of macrophages. Thus, WBP5 may impair the function of anti-tumorigenic macrophages, potentially exacerbating tumor progression (Figure 4D).
Overall, these findings underscore the multifaceted role of WBP5 in shaping the tumor immune microenvironment and its potential involvement in immune evasion. By influencing the infiltration and functionality of critical immune cell subsets, WBP5 may contribute to an immunosuppressive microenvironment that promotes tumor growth and resistance to immune-mediated therapies. This highlights the importance of further investigating WBP5 as a potential target for combination therapies aimed at enhancing antitumor immunity.
After; Page 6, line 174
The interplay between WBP5 expression and immune cell infiltration was evaluated to elucidate its potential role within the tumor microenvironment and its influence on immune-mediated mechanisms. Immune cell infiltration is a critical determinant of tumor progression, immune escape, and clinical outcomes, as infiltrating immune cells can exhibit either tumor-suppressive or tumor-promoting functions depending on their subtype and activation state. Deciphering the extent to which WBP5 expression modulates or correlates with the infiltration of specific immune cell populations is essential for understanding its contribution to the tumor-immune landscape and its potential utility as a therapeutic target in immunotherapy. Elevated WBP5 expression levels were strongly associated with significantly worse survival outcomes in patients exhibiting either increased or decreased levels of key immune cell subsets, including natural killer T (NKT) cells, regulatory T cells, CD4+ memory T cells, and macrophages (Figure 4). A high WBP5 expression level was associated with an increased risk of mortality in patients with enriched (HR = 2.54, p = 0.0052) and decreased NKT cells (HR = 1.66, p = 0.0096) (Figure 4A). Therefore, increased WBP5 expression levels could hinder the antitumor function of NKT cells, which are essential for identifying and destroying malignant cells.
A similar association was observed in regulatory T cells. In patients with enriched regulatory T cell populations, high WBP5 expression levels predicted poor survival (HR = 1.53, p = 0.044), whereas in those with decreased regulatory T cells, the HR was even higher (HR = 2.53, p = 0.0012). Thus, WBP5 may influence the immunosuppressive functions of regulatory T cells, thereby contributing to tumor immune evasion (Figure 4B). The relationship between WBP5 expression and CD4+ memory T cells was examined. A high WBP5 expression level was associated with worse outcomes in patients with both enriched (HR = 1.86, p = 0.0048) and decreased (HR = 2.05, p = 0.0025) CD4+ memory T cell levels, highlighting its broad impact on adaptive immune responses (Figure 4C).
In cancer, macrophages exhibit dual characteristics functioning as either pro-tumorigenic (M2-like) or antitumorigenic (M1-like) entities. When WBP5 was overexpressed, patient survival was reduced regardless of the presence or absence of macrophages. Thus, WBP5 may impair the function of anti-tumorigenic macrophages, potentially exacerbating tumor progression (Figure 4D).
Overall, these findings underscore the multifaceted role of WBP5 in shaping the tumor immune microenvironment and its potential involvement in immune evasion. By influencing the infiltration and functionality of critical immune cell subsets, WBP5 may contribute to an immunosuppressive microenvironment that promotes tumor growth and resistance to immune-mediated therapies. This highlights the importance of further investigating WBP5 as a potential target for combination therapies aimed at enhancing antitumor immunity.
- On figure 5, the WBP5 expression did not increase with the cancer stage, tumor grade and nodal metastasis status. It seems that WBP5 expression is not related to the cancer burden or severity. The authors should give more comprehensive explanation of the results of Figure 5.
Response:
Thank you for your thoughtful comment regarding Figure 5. While WBP5 expression does not show a strictly linear increase across all cancer stages, tumor grades, or nodal metastasis statuses, the results still reveal significant trends. For tumor stage (Figure 5A), WBP5 expression significantly increases from normal tissues to early-stage (Stage I) cancers and remains elevated in advanced stages, suggesting an early and sustained role in tumor progression. Similarly, for tumor grade (Figure 5B), WBP5 expression is higher in high-grade tumors, aligning with its association with aggressive tumor phenotypes. Regarding nodal metastasis (Figure 5C), WBP5 expression is elevated in metastatic cases (N1–N3) compared to non-metastatic cases (N0), highlighting its link to the metastatic phenotype. These findings suggest that WBP5 expression may reach a threshold early in tumor progression and contribute to the maintenance and aggressiveness of HNSCC. We have clarified these points in the revised results for a more nuanced interpretation. Thank you again for your valuable feedback.
Before; Page 8, line 208
Using tumor–node–metastasis (TNM) plot analysis, we investigated the association between WBP5 expression and critical clinical parameters, such as tumor stage, grade, and nodal metastasis status, to further elucidate the role of WBP5 in the progression and aggressiveness of HNSCC. WBP5 expression varied significantly across tumor stages, displaying a progressive increase in expression with advancing disease severity (Figure 5A).
Similarly, analysis of WBP5 expression based on tumor grade revealed a strong positive correlation between expression levels and tumor differentiation status (Figure 5B). Patients with high-grade tumors, characterized by poorly differentiated and aggressive cancer cells, exhibited significantly higher WBP5 expression levels than those with low-grade tumors.
Furthermore, WBP5 expression was examined in relation to nodal metastasis to assess its potential involvement in the metastatic process (Figure 5C). Patients with nodal metastasis, a key indicator of advanced disease and poor prognosis, exhibited significantly higher WBP5 expression levels than those without metastasis.
After; Page 8, line 228
Using tumor–node–metastasis (TNM) plot analysis, we investigated the association between WBP5 expression and critical clinical parameters, including tumor stage, tumor grade, and nodal metastasis status, to better understand its role in the progression and aggressiveness of head and neck squamous cell carcinoma (HNSCC).
First, the analysis of WBP5 expression across tumor stages revealed significant differences between normal tissues and cancerous tissues, with a marked increase observed in tumors (Figure 5A). While the expression did not show a strictly linear progression across all stages, it remained consistently elevated in advanced stages (Stage III and IV) compared to early-stage cancers (Stage I and II). This pattern suggests that WBP5 plays an important role in tumor initiation and continues to influence tumor progression. WBP5 expression was evaluated based on tumor grade, revealing a strong association with tumor differentiation status (Figure 5B). High-grade tumors, characterized by poorly differentiated and more aggressive cancer cells, exhibited significantly higher WBP5 expression levels compared to low-grade tumors and normal tissues. Although the differences between intermediate grades were less pronounced, the elevated expression in high-grade tumors highlights WBP5’s role in promoting aggressive tumor phenotypes. Lastly, we examined the relationship between WBP5 expression and nodal metastasis to assess its potential involvement in metastatic progression (Figure 5C). Tumors with nodal metastasis (N1–N3) showed significantly higher WBP5 expression levels compared to those without metastasis (N0) and normal tissues. Although expression levels did not significantly vary among metastatic groups (N1–N3), the overall elevation in metastatic cases indicates WBP5’s association with the metastatic phenotype, potentially contributing to the spread of HNSCC. Taken together, these results demonstrate significant correlations between increased WBP5 expression and advanced tumor stages, higher tumor grades, and nodal metastasis. Although the expression patterns are not strictly dose-dependent, WBP5 appears to play a critical role in tumor initiation, progression, and the development of aggressive and metastatic phenotypes. These findings position WBP5 as a promising prognostic biomarker and a potential therapeutic target for patients with advanced or high-risk HNSCC. Further studies are warranted to elucidate the molecular mechanisms underlying WBP5’s role in tumor progression and to explore its utility in therapeutic interventions.